# Electrophysiological Characteristics of Inhibitive Control for Adults with Different Physiological or Psychological Obesity

**DOI:** 10.3390/nu16091252

**Published:** 2024-04-23

**Authors:** Jiaqi Guo, Xiaofang Wan, Junwei Lian, Hanqing Ma, Debo Dong, Yong Liu, Jia Zhao

**Affiliations:** 1Faculty of Psychology, Southwest University, Chongqing 400715, China; swuguojiaqi@email.swu.edu.cn (J.G.); wanxiaofang@email.swu.edu.cn (X.W.); ljw981846236@email.swu.cn (J.L.); mahanqing@email.swu.edu.cn (H.M.); debo.dong@gmail.com (D.D.); 2Key Laboratory of Cognition and Personality (Ministry of Education), Southwest University, Chongqing 400715, China

**Keywords:** obesity, inhibitory control, working memory, negative-physical-self scale, electrophysiological features

## Abstract

Individuals exhibiting high scores on the fatness subscale of the negative-physical-self scale (NPSS-F) are characterized by heightened preoccupation with body fat accompanied by negative body image perceptions, often leading to excessive dieting behaviors. This demographic constitutes a considerable segment of the populace in China, even among those who are not obese. Nonetheless, scant empirical inquiries have delved into the behavioral and neurophysiological profiles of individuals possessing a healthy body mass index (BMI) alongside elevated NPSS-F scores. This study employed an experimental paradigm integrating go/no-go and one-back tasks to assess inhibitory control and working memory capacities concerning food-related stimuli across three adult cohorts: those with normal weight and low NPSS-F scores, those with normal weight and high NPSS-F scores, and individuals classified as obese. Experimental stimuli comprised high- and low-caloric-food pictures with concurrent electroencephalogram (EEG) and photoplethysmogram (PPG) recordings. Individuals characterized by high NPSS-F scores and normal weight exhibited distinctive electrophysiological responses compared to the other two cohorts, evident in event-related potential (ERP) components, theta and alpha band oscillations, and heart rate variability (HRV) patterns. In essence, the findings underscore alterations in electrophysiological reactivity among individuals possessing high NPSS-F scores and a healthy BMI in the context of food-related stimuli, underscoring the necessity for increased attention to this demographic alongside individuals affected by obesity.

## 1. Introduction

Executive function is one of the most related functions to obesity, which include working memory, inhibition control, and cognitive flexibility [1]. Among them, inhibition control should be the core ability that prevents people from becoming obese, as it plays a critical role in not overeating when people are facing fascinating food [2]. Working memory refers to a small amount of information that is held in mind temporarily and used in the execution of cognitive tasks, which inhibits us from losing the purpose of what we are doing [3,4]. Cognitive flexibility is the ability to shift one’s thinking from one task set to another task set [5]. All three abilities are critical factors in the formation of people’s balance between food intake and body consumption.

Many studies have found deficits in inhibitory control, working memory, and cognitive flexibility among people with obesity/overweight [4,6,7,8,9,10,11,12,13]. People with weak executive function have a higher risk of impulsive eating behavior when facing delicious food, and are more likely to focus on the food’s appearance and forget about satiety, causing an imbalance between their food intake and body composition [14,15], which finally leads to an increase in body weight. This might be one key reason for people becoming overweight or obese. Go/no-go and N-back tasks were frequently employed during the study of executive function in people with obesity [16]. It was found that the response accuracy of people with obesity was lower than that of people with normal weight when using low-calorie-vegetable pictures as the go condition and high-calorie cake as the no-go condition [16]. In an eye-motivation study, overweight or obese people paid more attention to food locations than people who were normal weight, indicating a typical attention bias to food [11]. Even though the target stimuli appeared, they could not resist seeing food stimuli. Food types are another factor that can influence an individual’s response [17]. They cannot inhibit themselves from high-calorie food and also cannot prevent the attraction to low-calorie food, but individuals with obesity perform similarly to those of normal weight on neutral and object stimuli. A meta-analysis found that people with obesity have lower execution ability than normal-weight people, with reduced inhibition control, working memory, and cognitive flexibility [4]. Besides reduced executive function, people with obesity also show reduced brain volume [18,19], morphometric similarity [20], and changed functional connectivity in the resting state [21,22].

Recently, “the thinner the beauty” has been popular in China, with many people continuing to lose weight even though they are not fat [7,23]. A group of people appeared to have a healthy BMI, but their negative-physical-self scale in the fatness subscale (NPSS-F) was high. By continually being on a diet, they were verified to have illiberal behavior in their eating behaviors and attention to food [24,25]. However, few studies are focused on their neurophysiological mechanisms, especially the differences in the electrophysiological features of people with obesity and people with normal weight and high NPSS-F. The above studies also divided the participants into two groups: the obese group and the normal-weight group, where the group of people with normal weight and high NPSS-F was often omitted. This might influence the accuracy of the results due to the special features of the NPSS-F group. Therefore, it is important to take people with high NPSS-F and normal weight into consideration, which may make the relationship between obesity and inhibitory control more clear.

The electroencephalogram (EEG) is a classical and popular technique due to its convenience and high cost-effectiveness. Voltage fluctuations and neural oscillations are typical features of the cognitive tasks measured by the EEG [26]. The P300, or its subcomponents P3a and P3b, was the key component in inhibitive control and working memory [27,28]. P3a has been involved in attention engagement and is often maximized at the position of frontal/central electrodes [29]. P3b was related to the cognitive workload for the experimental task [30]. Larger P3b will be elicited when the cognitive workload is higher, and it will be decreased when attention allocation occurs before it consumes more cognitive capacity. Individuals with schizophrenia, alcoholics, and some other substance users showed reduced P3b amplitude, indicating a deficit in working memory and information processing [31,32,33]. Theta rhythm is related to working memory and conflict control [34,35], and the alpha rhythm is associated with attention, working memory, and inhibitory control, respectively [36,37,38]. Studies of people with obesity found their executive functions are in deficit [9], but there are no studies about the executive functions of people with normal BMI and high NPSS-F.

Heart rate variability (HRV) has attracted much attention in obesity studies as it is related to neurocardiac function and the modulation of the autonomic nervous system (ANS), which has been a term in research terminology describing heart–brain interactions or heart–brain axis [39,40,41]. It was found that HRV was reduced in overweight/obese children [42]. HRV can predict EEG grade in infants with hypoxic conditions [43], affect the brain network related to emotion regulation [44], and modulate central neuronal activity [40,41]. Considering the strong correlation between brain and heart activities, it is interesting to study both of them to detect the effects of the heart–brain axis on obesity.

In recent years, the combination task of go/no-go and two-back paradigms has verified similar results to the two separate tasks [45]. As the combination task has the advantages of reducing time and avoiding the influence of task sequence and repetition during measurement, it should be significant to study executive function with the combination task. Considering that working memory and inhibitory control support one another and rarely appear independently [1], the above features inspired us to utilize food pictures with high and low calories as stimuli materials for the go/no-go and one-back combination task.

Taking the above together, this study explored the obese characteristics of people based on the combined task (go/no-go and one-back paradigms), high- and low-calorie food stimuli, and multimodal data (questionnaires, behavior, EEG, and HRV). It was hypothesized that (1) physiological obesity and negative-physical-self in fatness subscale will influence the amplitude of P3a, P3b, theta, and alpha during the combination task, resulting in lower P3a, P3b, theta, and alpha for people with physiological obesity (deficits in executive function) and higher P3a, alpha, and lower P3b and theta for people with normal BMI and high NPSS-F (excessive concern about their food intake); (2) people with obesity will have poorer HRV features than other groups; (3) the interaction between behavior, heart, and brain will be changed for people in the obese group and the high NPSS-F group compared with those in the normal group.

## 2. Materials and Methods

### 2.1. Participants

At the initiation of the study, a cohort of 60 participants was enrolled from Southwest University in Chongqing, evenly distributed into three groups with 20 participants per group, comprising 10 males and 10 females in each. Regrettably, due to inexperienced handling of the newly acquired EEG equipment, the EEG data of 6 participants were devoid of essential annotations pertaining to the occurrence of stimuli. Consequently, conducting event-related analyses on this subset of participants became unfeasible. Furthermore, subsequent assessment of participants’ BMI revealed that 3 individuals failed to meet the predetermined inclusion criteria whose BMIs were lower than 18.5 kg/m^2^. Consequently, the final analysis encompassed a sample size of 51 participants who fulfilled the necessary requirements for the study. Their ages ranged from 18 to 22 years old. They were healthy with normal or corrected-to-normal vision and had no neurological or psychological diseases. They were divided into three groups according to their body mass index (BMI) and negative-physical-self scale of fatness subscale (NPSS-F): that is, the group with obesity (BMI ≥ 25 kg/m^2^), the group with high NPSS-F and normal weight (BMI in [18.5, 23.9] kg/m^2^, NPSS-F > 2.5), and the group with normal NPSS-F and normal weight (BMI in [18.5, 23.9] kg/m^2^, NPSS-F < 1.5). The classification of BMI was based on the recent investigation and aligned with the established criteria for the adult population in China [8,46,47]. All the participants gave their written consent to take part in the experiment, and the study was approved by the Southwest University Ethics Committee.

### 2.2. Materials

#### 2.2.1. Self-Measurements

(1)Hunger, Thirst, and Desire to Eat

Participants rated their hunger, thirst, and desire to eat by selecting a value on a 100 mm visual analog scale (VAS), rated from “not at all” to “very high”. This was an important measure in this study because participants’ hunger states were strongly associated with their performance during the food-related task [6,7]. The scale included 3 questions, as follows: ① “What about your hunger degree now?”; ② “What about your thirst level now?”; ③ “How strong is your current desire to eat?”. The mean value of the eight questions was used as the index of eating desire for each participant.

(2)Dutch Eating Behavior Questionnaire at Restraint Subscale (DEBQ-RS)

The Dutch Eating Behavior Questionnaire has 33 items for assessing people’s eating behavior, which can be divided into 3 subscales, including emotional eating, external eating, and restrained eating [48]. Here, the restraint scale of the Dutch Eating Behavior Questionnaire with 10 items (e.g., “Do you take into account your weight with what you eat?”; “When your weight increases do you eat less than usual?”) was utilized to evaluate the participants’ diet behavior. Each item was rated on a 5-point scale from “never (1)” to “always (5)”, representing their restrained eating habits. In other words, participants with higher DEBQ-RS scores reflected more restrained eating and showed greater control over eating. The subscale has been verified to have high internal consistency across weight category groups. In a nonclinical sample of participants with normal weight, overweight, and obesity, Cronbach’s alpha coefficient ranged from 0.92 to 0.94 for the DEBQ-RS [49].

(3)Negative-Physical-Self Scale at Fatness Subscale (NPSS-F)

The NPSS fat subscale includes 11 items to measure affective, cognitive, and behavioral expressions of body satisfaction. It was developed by Chen Hong et al. in 2006, and has a Cronbach’s alpha coefficient of 0.88 [23]. Five-point scales were set for each item (e.g., “My weight has always been a pain in my heart.”; “I think I am fat in others’ eyes”), with an increasing trend from “never (0)” to “always (4)” representing their dissatisfaction with their body image. High NPSS-F scores indicate that participants are not satisfied with their body weight or consider themselves obese. In this study, people with scores of negative body image on the fatness subscale averaged more than 2.5 on the NPSS-F questionnaire. This was consistent with grouping criteria in previous studies [7,50].

(4)Positive Affect and Negative Affect Schedule (PANAS)

PANAS was developed by Watson et al. in 1988 to measure the emotional states of individuals at the current time [51]. It has 20 items and utilizes a 5-point scale for each item, with an increasing trend from “not at all (1)” to “extremely (5)” representing their emotion. It should be noted that positive and negative affects must be summed up separately.

#### 2.2.2. Experimental Procedure

A mixed experimental design was employed during the study, with 3 groups as between-group variables, food stimuli with high or low calories, and “go” or “no-go” requirements as within-group variables. Thus, a 3 × 2 × 2 experimental design was set. The food go/no-go task combined the typical go/no-go and one-back tasks. The task required the participant to respond to the food stimulus if the same stimulus was displayed adjacently before it, i.e., the last stimulus was the same as the current stimulus. Otherwise, the participant did not need to respond to food stimuli. The experimental procedure for one trial is shown in Figure 1. The task included two runs, and each run contained 120 trials. The ratio for go and no-go trials was 2:1: that is, 160 trials (80 for high-calorie food and 80 for low-calorie food) for go stimuli and 80 trials (40 for high-calorie food and 40 for low-calorie food) for no-go stimuli. One minute was set as the interval between the two runs for relaxation. Participants were required not to move and reduce blinks during the task. In each trial, a fixation was displayed for 500 ms, and then a food stimulus with high or low calories randomly appeared for 1000 ms. If the participant pressed the response key, the food stimulus disappeared immediately and a blank screen appeared. The whole duration of the task was around 10 min. Participants received about 60 Ұ as experimental reward.

The participants were randomly arranged to complete the experiment. Before they came to the laboratory, they needed to complete an NPSS-F questionnaire with sex and BMI reported simultaneously. When the participant met our requirements, she/he was informed to participate in the experiment. After receiving their informed consent, their height and weight was measured to calculate the BMI, ensuring that their reported BMI was correct. Their heart rate in a resting state was also recorded with PPG sensor in the NeXus-10 for 10 min while the participants were sitting on a chair while relaxing. Then, the food go/no-go task were carried out with their EEG recorded. After that, several questionnaires were collected from them, which included their hunger score, appetite, PFS, DEBQ-RS and PANAS.

#### 2.2.3. Data Recording and Analysis

(1)Behavior analyses

The mean (M) and standard deviation (SD) of self-report information were calculated for the three groups. Additionally, one-way analysis of variance (ANOVA) was used to test for group differences, including age, BMI, hunger level, thirst level, desire to eat, NPSS-F, DEBQ-RS, and PANAS. Further, for go_RT, go_ACC, and no-go_ACC, we conducted a 3 (group: overweight individuals, individuals with high NPSS-F and normal BMI, and individuals with normal weight) × 2 (stimulus: high-calorie food and low-calorie food) repeated measures analysis of variance, with group as the between-subjects factor and stimulus as the within-subjects factor. All analyses were conducted using SPSS software 26 (IBM, Armonk, NY, USA).

(2)Heart rate variability analyses

Heart rate was measured using a photoplethysmogram (PPG) sensor in Nexus-10 biotrace equipment. It was acquired before the experiment began with a 10 min resting state PPG signal. Participants were required not to move or think during this period. It was analyzed using the HRVTool toolbox with batch code including the baseline filtering and R peak value detection [52,53]. Before data analysis, the fluctuation of PPG data was visually examined, and abnormal fluctuation was manually removed. After the preprocessing, multiple HRV features were calculated which included the temporal domain (HR: the average heart rate; SDSD: the standard deviation of successive differences for heartbeat intervals; SDNN: the standard deviation of normal heartbeat intervals; RMSSD: the root square of successive heartbeat interval differences; pNN50: the percentage of successive heartbeat intervals that differ by more than 50 ms; TRI: the integral of the density of the heartbeat interval histogram divided by its height; TINN: the baseline width of the heartbeat histogram; rrHRV: the heart rate variability based on the relative heartbeat intervals), frequency domain (LF: the power of low frequency band (0.04–0.15 Hz); HF: the power of high frequency band (0.15–0.4 Hz); LHFratio: the ratio of power for low frequency band (0.04–0.15 Hz) and high frequency band (0.15–0.4 Hz); VLF: the power of very low frequency band (0.0033–0.04 Hz); pLF: the ratio of power of low frequency band (0.04–0.15 Hz) among the difference value of total power subtracting VLF; pHF: the power of high frequency band (0.15–0.4 Hz) among the difference value of total power subtracting VLF), and the nonlinear domain (CD: the correlation dimension; DFA: the detrended fluctuation analysis for short-term (DFA1) and long-term (DFA2); ApEn: the approximate entropy) [52,53].

(3)EEG analyses

EEG data were recorded with 64 Ag/AgCl scalp electrodes using the NeSen W series wireless EEG (Neuracle Technology Co., Ltd.; Changzhou, China). The sampling rate of the EEG equipment was 1000 Hz. EEG data were analyzed using EEGLAB v2022.0 (Swartz Center for Computational Neuroscience, La Jolla, CA, USA) [54], an open-source toolbox running on MATLAB software R2022a (The MathWorks, Inc., Natick, MA, USA). EEG data were filtered using a finite impulse response (FIR) filter with a bandpass of 1–30 Hz. Following that, EEG trials were extracted (−200 ms to 1000 ms) and visually inspected from trial to trial. EEG data with larger amplitudes or chaotic fluctuations were excluded from the initial trials. In the meantime, trials with response times outside of 250–1950 ms seemed half-hearted and were discarded. After that, independent component analysis (ICA) was applied to the remaining trials to remove eye blinks, head movements, and channel noise from the data for each participant [6,55]. After removing the independent components, the trials were inspected again to ensure no bad trials remained [7,55]. Then, the grand-averaged ERP difference was calculated for each group (obesity, NPSS-F, and control) by subtracting the ERP amplitude of the go condition from the no-go condition, with a reference baseline between −200 ms and 0 ms. Based on previous studies and our experimental design, P3a (310–380 ms) and P3b ERP components (440–510 ms) were compared between different groups and high- and low- calorie food stimuli.

Time–frequency analysis (TFA) with the short-time Fourier transformation (STFT) algorithm was used to calculate the power changes during this task, with a 200 ms window size and 50 ms step length to maintain a suitable tradeoff between time and frequency resolution [55,56]). Then, the baseline correction was applied to the TF results with the mean amplitude of each frequency in the pre-stimulus time interval (−200 ms to 0 ms) as the baseline, and this value was subtracted from the amplitude of its corresponding frequency in all the epochs, making the amplitude fluctuation of frequencies after the onset of stimulus relative values. Finally, the grand-averaged TF results were calculated for each electrode under four kinds of selections, and the regions of interest (ROI) in the alpha (8–10 Hz, 0.3–0.9 s) and theta (4–7 Hz, 0.3–0.9 s) ranges were extracted and compared between different selections. Three electrodes distributed in the frontal cortex (F3, Fz, and F4) were selected during the statistical process.

Finally, SPSS 27 was utilized for the statistics process, where a three-way repeated-measures ANOVA was applied to the mean amplitudes of P3a, P3b, theta rhythm, and alpha rhythm, respectively, resulting in a 3 (groups of subjects) × 2 (food types: high-calorie/low-calorie) × 3 (position of electrodes: F3/Fz/F4) ANOVA for each component. Finally, either the post-doc analysis or a simple effects analysis was also tested with SPSS, according to the ANOVA results.

(4)Correlation analysis

Pearson correlation was applied to identify if heart–brain axis changes existed or if the interaction between questionnaires and electrophysiology was changed. Therefore, 9 behavioral features, 12 ERP features, 24 power features (12 theta and 12 alpha), and 18 HRV features were brought into the calculation. After finding the strong in-class correlation, a between-class correlation was calculated again to clearly show the results. Only the r value with *p* < 0.05 is shown in the results below.

## 3. Results

### 3.1. Self-Reported Results

One-variable ANOVA was applied to each variable, and the results are shown in Table 1. It was found that there were significant differences in DEBQ-RS, NPSS-F, and BMI, while no other differences were found. Post-doc tests with LSD correction found that the obese group had a higher BMI than the control group (*p* < 0.001) and the NPSS-F group (*p* < 0.001). The control group had lower NPSS-F than the NPSS-F group (*p* < 0.001) and the obese group (*p* < 0.001). The DEBQ-RS of the control group was also lower than that of the NPSS-F group (*p* < 0.001) and the obese group (*p* = 0.002). No other significant results were found during the post-doc tests.

### 3.2. HRV Results

One-variable ANOVA was applied to each HRV feature, and the significant results for groups were found in pLF (F = 4.053, *p* = 0.024), pHF (F = 4.053, *p* = 0.024), and LHF ratio (F = 4.252, *p* = 0.020). Post-doc tests with LSD correction showed that pLF of the normal group was much larger than that of the NPSS-F group (*p* = 0.041) and the obese group (*p* = 0.009), pHF of the normal group was much smaller than that of the NPSS-F group (*p* = 0.041) and the obese group (*p* = 0.009), and LHF ratio of the normal group was much larger than that of NPSS-F group (*p* = 0.021) and obese group (*p* = 0.01).

### 3.3. EEG Results

The ERPs and time–frequency results are shown in Figure 2 and Figure 3, respectively. 2 (low- vs. high-calorie food) × 3(normal/obesity/npss) × 3(F3/Fz/F4) repeated-measurement ANOVAs were applied to the amplitudes of P3a, P3b, theta rhythms, and alpha rhythms. It was found that the main effects for electrodes were significant (F = 16.398, *p* < 0.001; η^2^ = 0.255) and the interaction effects between groups and food stimuli were significant (F = 3.411, *p* = 0.041, η^2^ = 0.124). Post-doc tests found that the amplitude of P3a at Fz was much larger than that of F3 (*p* < 0.001) and F4 (*p* < 0.001). The simple-effects test showed that the P3a amplitude under high-calorie food in the NPSS-F group was much larger than that under low-calorie food (*p* = 0.043), and the P3a amplitude in the control group under low-calorie food was larger (marginally significant) than that in the NPSS-F group (*p* = 0.077) and the obese group (*p* = 0.058).

Repeated-measurement ANOVA for the amplitude of P3b found that the main effects between electrodes were significant (F = 18.786, *p* < 0.001, η^2^ = 0.281), and the main effect for food stimuli was significant (F = 5.341, *p* = 0.025; η^2^ = 0.1). Post-doc tests found that the P3b amplitude at F3 was larger than that of Fz (*p* < 0.001) and F4 (*p* = 0.012), and the P3b amplitude at Fz was smaller than that at F4 (*p* = 0.003). Post-doc tests for food stimuli showed that high-calorie food can induce a larger P3b amplitude than low-calorie food (*p* = 0.025).

Repeated-measurement ANOVA for the amplitude of theta rhythm found that the main effects for groups (F = 4.493, *p* = 0.016; η^2^ = 0.158) and electrodes (F = 14.582, *p* < 0.001, η^2^ = 0.233) were significant. The post-doc tests found that the theta power of the NPSS group was much larger than that of the normal group (*p* = 0.016) and the obese group (*p* = 0.010), and the theta power at F3 was larger than that at Fz (*p* < 0.001) and F4 (*p* < 0.001).

Repeated-measurement ANOVA for the amplitude of alpha rhythm found that the main effect for electrodes was significant (F = 4.024, *p* = 0.024; η^2^ = 0.077), the main effect for groups was marginally significant (F = 2.858, *p* = 0.067; η^2^ = 0.106), and the interaction effect between electrodes, food stimuli, and groups was significant (F = 2.96, *p* = 0.035; η^2^ = 0.11). The post-doc test showed that the alpha power of the NPSS-F group was higher than that of the obese group (*p* = 0.023), and the alpha power at F3 was higher than that at Fz (*p* = 0.008) and F4 (*p* = 0.02). A simple-effects test found that the alpha power for high-calorie food in the NPSS-F group was higher than that of the obese group at F3 (*p* = 0.008) and F4 (*p* = 0.003).

The correlation results are shown in Figure 4. Figure 4a is the significant results of Pearson correlation between all variables (*p* < 0.05), and Figure 4b is the significant results of Pearson correlation between variables from different subsets: that is, the correlation results in a subset were removed. In Figure 4a, it can be found that more correlations are found in subsets than the correlation between subsets. To show the results more clearly, we removed the inner correlation of each subset and plot the results in Figure 4b. It can be found that the most stable correlation is the relationship between NPSS and hunger. For the other correlations, most of them are different between three groups.

## 4. Discussion

Based on the aforementioned findings, the study revealed a significant association between the amplitude of the P3a component (i.e., the difference in amplitude between go and no-go stimuli) and participants’ responses to high- and low-calorie food stimuli. Specifically, the NPSS-F group exhibited a significantly greater P3a amplitude in response to high-calorie food stimuli compared to low-calorie food stimuli. This observation suggests that the NPSS-F group demonstrates heightened attention towards the caloric content of food stimuli due to their daily abstinence from food intake. Furthermore, disparities among the groups were also evident in their responses to low-calorie food stimuli, with both the NPSS-F and obese groups exhibiting smaller P3a amplitudes compared to the control group. These findings indicate that the NPSS-F and obese groups allocate less attention to low-calorie food stimuli compared to the control group. These differential food attractions likely stem from the participants’ food interactions, as the NPSS-F group exhibits greater restraint in consuming high-calorie food, resulting in increased desire for high-calorie options. P3b might be more related to decision-making as its amplitude was much larger under high-calorie food than low-calorie food. The distinct meanings attributed to identical food stimuli among the three groups further influence their P3b amplitudes [57]. When the high-calorie food appeared, it attracted their attention more than the low-calorie food. The special response to high-calorie food also led to the amplitude reduction of low-calorie food in the NPSS-F group. For the obese group, their response to food stimuli was a deficit, which led to a lower P3a amplitude. This might be the reason that the amplitude of P3a in the control group was the largest. Notably, the amplitude of P3b successfully discriminates between experimental conditions for all three groups, albeit high-calorie food stimuli pose greater challenges in differentiation from low-calorie stimuli. This indicates that obese individuals encounter difficulties in processing high-calorie food stimuli, potentially contributing to heightened cravings and unhealthy dietary choices.

EEG oscillations are known to be indicative of inhibitory control and working memory. Specially, theta rhythms have proven useful in discerning the NPSS-F group, control group, and the obese group, irrespective of the type of stimulus employed. Theta activity in the NPSS-F group exhibited the largest magnitude, suggesting heightened sensitivity to food stimuli in this group. Furthermore, alpha rhythms in the NPSS-F group were higher compared to the obese group, particularly in the context of high-calorie food stimuli.

In addition to EEG findings, significant alterations in heart rate variability (HRV) measures were observed among the obese and NPSS-F groups. Specifically, there was a notable decrease in the proportion of low-frequency oscillations (pLF) and a decrease in the LF/HF ratio, along with an increase in the proportion of high-frequency oscillations (pHF). These HRV patterns further contribute to our understanding of the mechanisms underlying obesity. Considering the multiple findings obtained across different groups, this study holds significance in advancing our knowledge of obesity mechanisms.

### 4.1. Inhibitive Control and Working Memory of Obesity

Executive function encompasses crucial cognitive abilities such as inhibitive control and working memory, which are known to interact with each other [2]. Deficits in executive function among individuals with obesity have been extensively documented through comparisons with normal-weight counterparts [4,8,58,59]. However, studies examining inhibitive control and working memory in individuals with both a healthy BMI and high NPSS-F scores are lacking, potentially introducing confounding factors and influencing study outcomes due to the confusion of the NPSS-F individuals with the control individuals. Consequently, this study aimed to differentiate the NPSS-F group from the control and obese groups based on multimodal features. Considering the NPSS-F group is imperative in future obesity studies.

Consistent with previous research that employed go/no-go tasks with English letters, it was observed that individuals with obesity exhibited reduced P300 amplitudes compared to normal-weight individuals [60]. This finding aligns with the current study, despite the stimuli differing from English letters to Chinese-food pictures. In this study, by including the NPSS-F group and increasing task difficulty through a combination of go/no-go and one-back tasks, the results aligned more closely with our initial hypotheses. Furthermore, the influence of food types [17] on P3a and P3b amplitudes among groups was also confirmed.

Given the distinct performances of the three groups, the dual process model is applicable to elucidate these findings [13]. The impulsive and control systems collaboratively determine behavioral performance. When individuals encounter highly tempting food cues, particularly high-calorie options, their level of control system activation influences their food intake. Those with low control abilities might frequently engage in overeating behaviors. Consequently, when faced with food cues or high-calorie options, individuals with lower control abilities might struggle to inhibit impulsive reactions, leading to behaviors strongly associated with obesity, such as overeating and weight gain. These theoretical explanations align well with the current study’s results.

In the current “obesogenic” environment, characterized by abundant food cues and easy access to various foods, especially high-calorie options, the sales of junk food have surged. Furthermore, the fat content in low-nutrient, high-calorie foods is challenging to offset through exercise [61]. Additionally, evolutionary psychology posits that humans naturally exhibit a preference for high-calorie foods, particularly those rich in fat and sugar [62]. Obese individuals tend to demonstrate a rapid approach behavior rather than avoidance behavior towards food-related stimuli [63]. Similarly, individuals with negative body self-schema exhibit attentional biases towards body shape or food-related information. This cognitive bias facilitates and expedites the information processing related to food and self-schema, resulting in selective preferences at different stages of information processing. Further exploration is required to determine whether attention bias towards negative information impacts normal brain activity, leading individuals with high NPSS-F scores to excessively attend to negative information and engage in more self-restraint behaviors during food selections. Alternatively, it is worth investigating whether excessive attention to negative information depletes cognitive resources, thereby weakening executive function. Diverging from the obese group, the NPSS-F group is less prone to overconsumption of high-calorie food due to biased self-perception. Individuals in the NPSS-F group proactively restrict their diet, reduce, or even reject high-calorie foods, indicating stronger self-control abilities and special cognitive bias to food and body fat. Therefore, it is speculated that individuals in the NPSS-F group exhibit superior inhibitory control and higher executive function capabilities, together with higher cognitive consumption when facing the food cues.

### 4.2. Brain–Heart Axis

The concept of the brain-heart axis initially focused on studying the effects of neurologic injuries on cardiovascular function [64]. However, more recently, it has been employed to explore the bidirectional interaction between the brain and the heart, as evidenced by the influence of heart activity on the central nervous system [41,42]. Given the strong correlation between obesity and cardiovascular diseases, it is crucial to investigate heart rate variability (HRV) alongside EEGs, as it may provide additional insights into the brain–heart axis. In the present study, HRV indices such as pLF and LHF ratio were significantly lower in the NPSS-F group and obese group compared to the control group. Conversely, pHF was substantially larger in these two groups than in the control group, which is consistent with previous studies [65,66]. The pLF and pHF are associated with the sympathetic nervous system (SNS) and the parasympathetic nervous system (PNS), respectively [65]. The ratio of LF and HF is considered a measure of autonomic nervous system (ANS) activity, with a high LHF ratio indicating increased SNS activity [65,67]. One study found a negative correlation between HRV (LF and LHF ratio) and participants’ body fat percentage [66]. The presence of both excitatory sympathetic and inhibitory parasympathetic activities is indicative of positive self-regulation and effective adaption [65].

### 4.3. Limitations and Future Directions

To the best of our knowledge, this study is innovative in its utilization of a novel experimental paradigm, multimodal measurements, new subject grouping, and multiple analysis methods. However, the relatively small number of participants is a limitation that should be addressed in future studies. The scarcity of obese individuals among Southwest University undergraduates posed challenges in recruiting obese participants. Therefore, it is crucial to expand the participant pool and increase the sample size in future studies to enhance the reliability of findings related to obesity. Additionally, the posture of participants during photoplethysmography (PPG) measurements should be taken into account in subsequent studies, as posture has been shown to influence HRV through different autonomic nervous system responses [68,69]. Whenever feasible, it is advisable to have participants lie in a supine position during measurements, as supine posture has been associated with relaxation and resting of the autonomic nervous system [70].

## 5. Conclusions

In summary, our study has successfully validated a substantial portion of the hypotheses. Firstly, notable differences were observed in the EEG indices between the NPSS-F group, the obese group, and the control group. These differences were reflected in reduced amplitude of P3a in both the NPSS-F and obese groups under low-calorie food stimuli, increased amplitude of theta power in the NPSS-F group compared to the normal and obese groups, and increased amplitude of alpha power in the NPSS-F group compared to the obese group.

Secondly, the analysis of HRV indices revealed distinct patterns in the NPSS-F and obese groups when compared to the control group. Specifically, there were decreased ratios of low-frequency (pLF) and low-to-high frequency (LHF ratio), as well as an increased high-frequency ratio (pHF) in both the NPSS-F and obese groups.

Furthermore, the correlation analysis among the behavioral, HRV, and EEG indices demonstrated varying interaction patterns among these variables across the three groups. These findings collectively indicate the potential utility of questionnaires, EEGs, PPGs, and experimental tasks in discerning the cognitive processing characteristics of the three groups. It also hold significant implications for future investigations aimed at deciphering the cognitive processes among different groups associated with inhibitory control and working memory.

## Figures and Tables

**Figure 1 nutrients-16-01252-f001:**
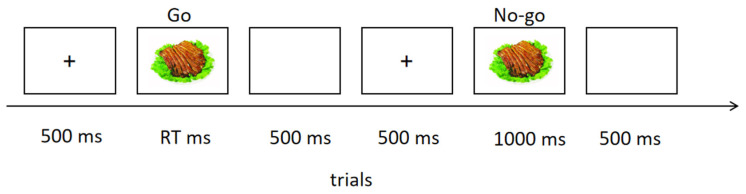
Experimental flowchart for the combination task of go/no-go and one-back paradigms. RT is the abbreviation of response time. When the stimulus is the same as the last stimulus before it, the participant is required not to respond to the stimulus, i.e., the no-go condition.

**Figure 2 nutrients-16-01252-f002:**
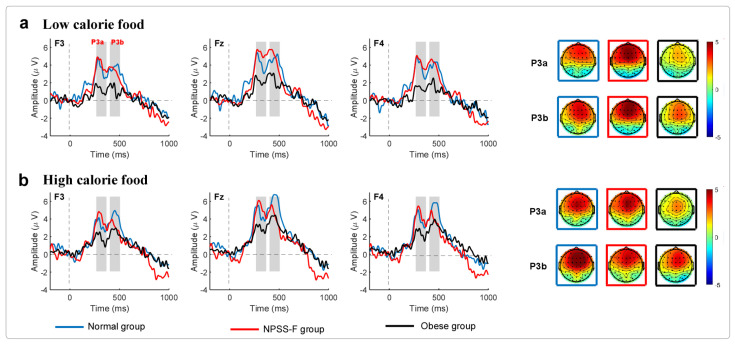
Event-related potentials (ERPs) and scalp topography for the three groups under high- and low-calorie food stimuli. The ERPs were the subtraction between no-go and go conditions, which suggests the inhibitory control ability.

**Figure 3 nutrients-16-01252-f003:**
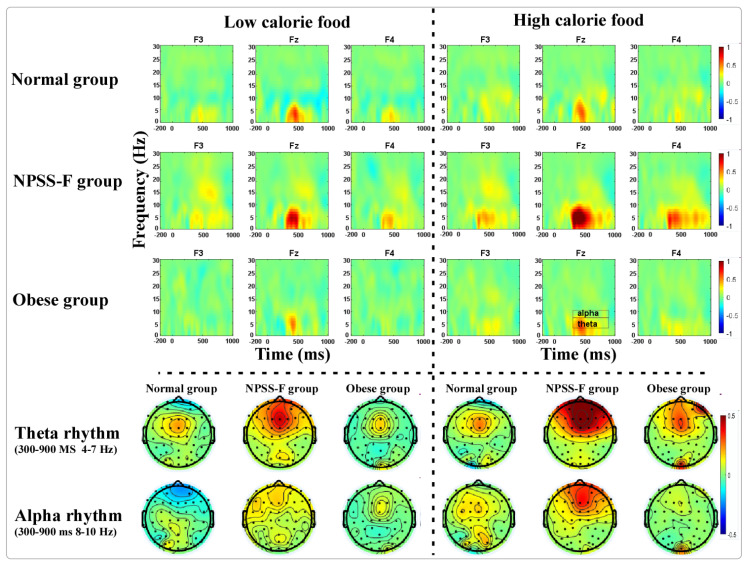
Event-related synchronization (ERS) of theta and alpha rhythms for three groups under high- and low-calorie food stimuli.

**Figure 4 nutrients-16-01252-f004:**
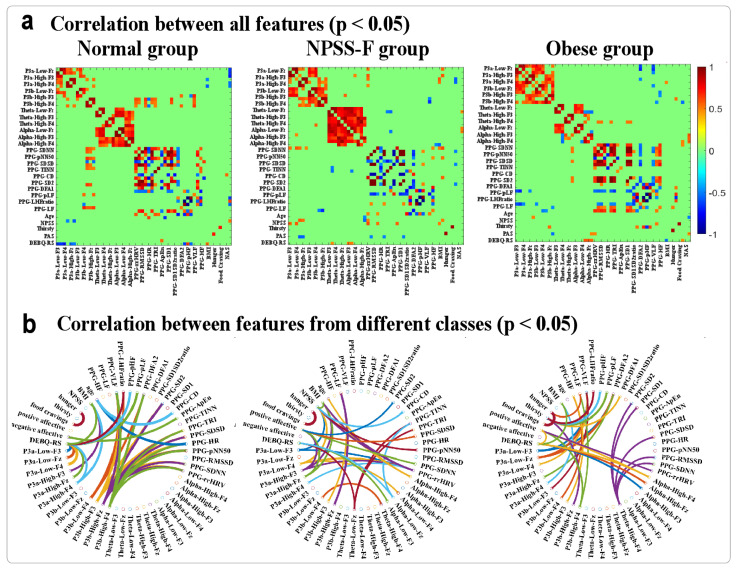
Correlation between all features (**a**) *p* < 0.05 and features from different subsets (**b**). In (**b**), the correlations between features of the same subset were removed to highlight the results between different subsets.

**Table 1 nutrients-16-01252-t001:** Demographic information and self-reported results.

Variable	Control Group (M ± SD) N = 16	NPSS-F Group (M ± SD) N = 17	Obese Group (M ± SD) N = 18	F	*p*
Age	19.38 (1.67)	19.82 (1.47)	19.78 (1.35)	0.448	0.64
DEBQ-RS ***	2.41 (0.81)	3.69 (0.68)	3.30 (0.82)	11.86	<0.001
NPSS-F ***	1.31 (0.21)	2.96 (0.33)	2.97 (0.38)	151.6	<0.001
Hunger	20.63 (16.52)	32.35 (15.62)	27.22 (17.76)	2.04	0.14
Thirst	41.25 (18.57)	36.47 (10.57)	32.22 (13.09)	1.68	0.198
Desire to eat	25.00 (18.26)	30.59 (17.13)	24.44 (19.17)	0.594	0.56
PAS	2.98 (0.73)	2.72 (0.58)	3.00 (0.61)	0.969	0.387
NAS	1.84 (0.72)	1.96 (0.70)	1.86 (0.63)	0.149	0.862
BMI ***	20.30 (1.38)	21. 40 (1.44)	26.82 (1.88)	83.13	<0.001

Note: *** *p* < 0.001. PAS: positive affective schedule; NAS: negative affective schedule.

## Data Availability

The data presented in this study are available on request from the corresponding author. The data are not publicly available due to concerns about privacy and ethics in personal decision-making.

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
