# Peer review of "Electrophysiological Characteristics of Inhibitive Control for Adults with Different Physiological or Psychological Obesity"

_nutrients, 2024, doi:10.3390/nu16091252_

Round 1
Reviewer 1 Report
Comments and Suggestions for Authors
1. What is the main question addressed by the research?
The work concerns the electrophysiological characteristics of inhibitory control in adults with various physiological or mental obesity. Although the results of many studies indicate deficits in inhibitory control, working memory and cognitive flexibility in obese/overweight people, the authors attempted to answer the question: What is the process of inhibitory control and working memory skills in relation to food-related stimuli in people adults with normal weight and low scores on the Negative-Physical-Self Scale in the Fatness (NPSS-F), with normal weight and high scores on the NPSS-F scale, and people classified as obese.
2. Do you consider the topic original or relevant in the field?
Based on literature data, the authors comprehensively characterized the role of working memory, inhibitory control and cognitive flexibility as critical factors in shaping the balance between food intake and body consumption. In the research, the authors experimentally verified three hypotheses stating that: 1. Physiological and psychological obesity will influence the amplitude of P3a, P3b, theta, and alpha during the combination task, resulting in lower P3a, P3b, theta, and alpha for people with physiological obesity (deficits in executive function) and higher P3a, alpha, and lower P3b and theta for people with normal BMI and high NPSS-F (excessive concern about their food intake); 2. people with obesity have poorer HRV features than other groups; 3. The interaction between behavior, heart, and brain was changed for people in the obese group and the high NPSS-F group.
3. What does it add to the subject area compared with other published material?
Although the topic of inhibitory control, working memory and cognitive flexibility in obese/overweight people as critical factors in shaping the balance between food intake and body consumption is quite often discussed in research works, there are few studies focusing on neurophysiological mechanisms, especially electrophysiological features in relation to people with obesity and people with normal body weight and healthy NPSS-F. Therefore, according to the reviewer, the results obtained in the study are something new, especially for the Chinese population.
4. What specific improvements should the authors consider regarding the methodology? What further controls should be considered?
The positive sides of the work are: the innovativeness of the study in terms of the use of a new experimental paradigm of multimodal measurements, a new grouping of topics and many analysis methods. The disadvantages of the work include the too small number of people participating in the research, especially since it was a group selected in terms of age and level of education. According to the reviewer, it would be advisable to conduct similar research on a larger population in different age groups and with different levels of education, performing different professions and living in different regions of China.
5. Are the conclusions consistent with the evidence and arguments presented and do they address the main question posed?
The conclusion presented in the study is very general and only indicates the existence of differences in physiological characteristics between the studied groups of students. According to the reviewer, it would be necessary to define the possible benefits resulting from the research conducted and the results obtained.
6. Are the references appropriate?
The literature is numerous, contemporary, well selected and correctly cited in the text of the publication.
7. Please include any additional comments on the tables and figures.
The table and charts presented in the work are clear and clearly described.
Author Response
Responds to the comments of Reviewer 1:
Reviewer 1:
- What is the main question addressed by the research?
The work concerns the electrophysiological characteristics of inhibitory control in adults with various physiological or mental obesity. Although the results of many studies indicate deficits in inhibitory control, working memory and cognitive flexibility in obese/overweight people, the authors attempted to answer the question: What is the process of inhibitory control and working memory skills in relation to food-related stimuli in people adults with normal weight and low scores on the Negative-Physical-Self Scale in the Fatness (NPSS-F), with normal weight and high scores on the NPSS-F scale, and people classified as obese.
Response: Thank you very much for your comments.
- Do you consider the topic original or relevant in the field?
Based on literature data, the authors comprehensively characterized the role of working memory, inhibitory control and cognitive flexibility as critical factors in shaping the balance between food intake and body consumption. In the research, the authors experimentally verified three hypotheses stating that: 1. Physiological and psychological obesity will influence the amplitude of P3a, P3b, theta, and alpha during the combination task, resulting in lower P3a, P3b, theta, and alpha for people with physiological obesity (deficits in executive function) and higher P3a, alpha, and lower P3b and theta for people with normal BMI and high NPSS-F (excessive concern about their food intake); 2. people with obesity have poorer HRV features than other groups; 3. The interaction between behavior, heart, and brain was changed for people in the obese group and the high NPSS-F group.
Response: Thank you sincerely for your thorough review of our manuscript.
- What does it add to the subject area compared with other published material?
Although the topic of inhibitory control, working memory and cognitive flexibility in obese/overweight people as critical factors in shaping the balance between food intake and body consumption is quite often discussed in research works, there are few studies focusing on neurophysiological mechanisms, especially electrophysiological features in relation to people with obesity and people with normal body weight and healthy NPSS-F. Therefore, according to the reviewer, the results obtained in the study are something new, especially for the Chinese population.
Response: Thank you for your valuable comments.
- What specific improvements should the authors consider regarding the methodology? What further controls should be considered?
The positive sides of the work are: the innovativeness of the study in terms of the use of a new experimental paradigm of multimodal measurements, a new grouping of topics and many analysis methods. The disadvantages of the work include the too small number of people participating in the research, especially since it was a group selected in terms of age and level of education. According to the reviewer, it would be advisable to conduct similar research on a larger population in different age groups and with different levels of education, performing different professions and living in different regions of China.
Response: Thank you very much for the valuable comments, which are significant for our future study. Indeed, we have the same idea and have prepared to establish a comprehensive database integrating multimodal equipment and conducting multiple experiments involving a substantial population. This approach holds promising prospects for unraveling extensive insights into the cognitive function of individuals with obesity. However, the execution of such an endeavor presents formidable challenges, primarily stemming from the considerable financial requirements and workload demands. Currently, our progress is gradual as we incrementally accumulate data, with the anticipation that the pace will expedite once additional funding becomes available. The limitation of this study with small sample size has also been stated in the discussion part.
- Are the conclusions consistent with the evidence and arguments presented and do they address the main question posed?
The conclusion presented in the study is very general and only indicates the existence of differences in physiological characteristics between the studied groups of students. According to the reviewer, it would be necessary to define the possible benefits resulting from the research conducted and the results obtained.
Response: Thank you very much for the valuable comments. We have added the illustration about the possible benefits resulting from the research here as the following: In summary, our study has successfully validated a substantial portion of the hypotheses. Firstly, notable differences were observed in the EEG indices between the NPSS-F, the obese groups, and the control group. These differences were reflected in reduced amplitude of P3a in both the NPSS-F and obese groups under low-calorie food stimuli, increased amplitude of theta power in the NPSS-F group compared to the normal and obese groups, and increased amplitude of alpha power in the NPSS-F group compared to the obese group.
Secondly, the analysis of HRV indices revealed distinct patterns in the NPSS-F and obese groups when compared to the control group. Specifically, there were decreased ratios of low-frequency (pLF) and low-to-high frequency (LHFratio), as well as in-creased high-frequency ratio (pHF) in both the NPSS-F and obese groups.
Furthermore, the correlation analysis among the behavioral, HRV, and EEG indices demonstrated varying interaction patterns among these variables across the three groups. These findings collectively indicate the potential utility of questionnaires, EEG, PPG, and experimental tasks in discerning the cognitive processing characteristics of the three groups. It also hold significant implications for future investigations aimed at deciphering the cognitive processes associated with inhibitory control and working memory.
- Are the references appropriate?
The literature is numerous, contemporary, well selected and correctly cited in the text of the publication.
Response: Thank you very much for the comments.
- Please include any additional comments on the tables and figures.
The table and charts presented in the work are clear and clearly described. Title is appealing, however the terminology psychologically obese should, in my opinion, be rethought.
Response: Thank you for your comments. We have revised the terminology psychologically obese as people with high NPSS-F and normal BMI.
Reviewer 2 Report
Comments and Suggestions for Authors
This is a quite interesting study on the electrophysiological characteristics of physiologically and psychologically obese individuals. It describes quite interesting phenomena that add to the knowledge of understanding the tendency to become overweight and obese in the future. The study is worth publishing, but the authors should revise their paper because of some serious errors and lack of clarity.
Line 122
You should state the initial number of volunteers who wanted to take part in your study, unless you were extremely lucky to have 51 participants and divided them into three fairly equal groups.
Lines 126-129
You need to give references to this BMI classification (I'm not convinced it's correct, isn't the normal BMI for the Asian-Pacific population between 18.5 and 22.9?) If I'm right, you need to recalculate your results. Remember that this paper, if accepted, will be read in different countries around the world and it will not be clear to all that the Asian BMI classification is different from the Caucasian BMI classification.
Your conclusions should refer to the hypothesis presented in the introduction. Also, the statement that "we need to help [people with high scores on the Negative-Physical-Self Scale in the Fatness Subscale] to be normal by changing their attention to food and their eating behaviour" is quite naive, because how would it be possible to find out who is NPSS-F without sophisticated analysis? You should delete it or at least rephrase it and explain what you're thinking.
Explain the abbreviations in Table 1, especially what is PAS, NAS, etc.
You need to improve the quality of Figure 2, 3 and especially Figure 4, which is completely unreadable.
Author Response
Responds to the comments of Reviewer 2:
Reviewer 2:
- This is a quite interesting study on the electrophysiological characteristics of physiologically and psychologically obese individuals. It describes quite interesting phenomena that add to the knowledge of understanding the tendency to become overweight and obese in the future. The study is worth publishing, but the authors should revise their paper because of some serious errors and lack of clarity.
Response: Thank you very much for your affirmation about the results of our manuscript and the comments to improve the quality of the study. We have revised/corrected the manuscript for the errors and language description. The changed parts are marked with blue color in the revised manuscript.
- Line 122:
You should state the initial number of volunteers who wanted to take part in your study, unless you were extremely lucky to have 51 participants and divided them into three fairly equal groups.
Response: Thank you for your comments. At the initiation of the study, a cohort of 60 participants was enrolled, evenly distributed into three groups with 20 participants per group, comprising 10 males and 10 females in each. Regrettably, due to inexperienced handling of the newly acquired EEG equipment, the EEG data of 6 participants were devoid of essential annotations pertaining to the occurrence of stimuli. Consequently, conducting event-related analyses on this subset of participants became unfeasible. Furthermore, subsequent assessment of participants’ BMI revealed that 3 individuals failed to meet the predetermined inclusion criteria. Consequently, the final analysis encompassed a sample size of 51 participants who fulfilled the necessary requirements for the study. We have added the illustration in “2.1 participants” part of the revised manuscript.
- Lines 126-129
You need to give references to this BMI classification (I'm not convinced it's correct, isn't the normal BMI for the Asian-Pacific population between 18.5 and 22.9?) If I'm right, you need to recalculate your results. Remember that this paper, if accepted, will be read in different countries around the world and it will not be clear to all that the Asian BMI classification is different from the Caucasian BMI classification.
Response: Thank you for your comments. In the previous years, normal BMI for the Asian-Pacific population was between 18.5 and 22.9 kg/m2 due to the relatively small body size. But in recent years, normal BMI for Chinese adults was defined as 18.5 ≤ BMI < 24 kg/m2 according to the national health commission of the People’s Republic of China (the report in 2024 states the BMI distribution as the following table). There are also studies with big population around different regions of China, which stated clearly the normal BMI range is [18.5 24). We have added the following two references in the revised manuscript.
Reference 1: Pan, X. F., et al. (2021). Epidemiology and determinants of obesity in China. Lancet Diabetes & Endocrinology, 9(6), 373-392.
Reference 2: Chen, K., et al. (2023). Prevalence of obesity and associated complications in China: A cross-sectional, real-world study in 15.8 million adults. Diabetes, Obesity and Metabolism, 25(11), 3390-3399.
Reference 3: Cui, K., et al. (2023). Higher visceral adipose tissue is associated with decreased memory suppression ability on food-related thoughts: A 1-year prospective ERP study. Appetite, 191, 107048.
- Your conclusions should refer to the hypothesis presented in the introduction. Also, the statement that "we need to help [people with high scores on the Negative-Physical-Self Scale in the Fatness Subscale] to be normal by changing their attention to food and their eating behaviour" is quite naive, because how would it be possible to find out who is NPSS-F without sophisticated analysis? You should delete it or at least rephrase it and explain what you're thinking.
Response: Thank you for your comments. We have delete this statement and make connection between the conclusion and hypothesis.
- Explain the abbreviations in Table 1, especially what is PAS, NAS, etc.
You need to improve the quality of Figure 2, 3 and especially Figure 4, which is completely unreadable.
Response: Thank you for your comments. We have added the explanation of the abbreviations under Table 1. PAS and NAS are the abbreviations of positive affective schedule and negative affective schedule, respectively.
Round 2
Reviewer 2 Report
Comments and Suggestions for Authors
Still, Figure 4 is barely readable.
Author Response
Thank you and apologize for this neglect. We have replotted Figure 4 to improve its clarity and unify the group names in Figures 2-4.